# Downregulation of IL-8 and IL-10 by LRRC8A Inhibition through the NOX2–Nrf2–CEBPB Transcriptional Axis in THP-1-Derived M_2_ Macrophages

**DOI:** 10.3390/ijms25179612

**Published:** 2024-09-05

**Authors:** Miki Matsui, Junko Kajikuri, Hiroaki Kito, Elghareeb E. Elboray, Takayoshi Suzuki, Susumu Ohya

**Affiliations:** 1Department of Pharmacology, Graduate School of Medical Sciences, Nagoya City University, Nagoya 467-8601, Japan; c241739@ed.nagoya-cu.ac.jp (M.M.); kajikuri@med.nagoya-cu.ac.jp (J.K.); kito@med.nagoya-cu.ac.jp (H.K.); 2Department of Complex Molecular Chemistry, SANKEN, Osaka University, Osaka 560-0043, Japan; e.elboray@sci.svu.edu.eg (E.E.E.); tkyssuzuki@sanken.osaka-u.ac.jp (T.S.); 3Department of Chemistry, Faculty of Science, South Valley University, Qena 83523, Egypt

**Keywords:** tumor-associated macrophage, LRRC8A, Cl^−^ channel, IL-8, IL-10, NOX2, Nrf2, CEBPB, epigenetic modification, tumor microenvironment

## Abstract

M_2_-polarized, tumor-associated macrophages (TAMs) produce pro-tumorigenic and angiogenic mediators, such as interleukin-8 (IL-8) and IL-10. Leucine-rich repeat-containing protein 8 members (LRRC8s) form volume-regulated anion channels and play an important role in macrophage functions by regulating cytokine and chemokine production. We herein examined the role of LRRC8A in IL-8 and IL-10 expression in THP-1-differentiated M_2_-like macrophages (M_2_-MACs), which are a useful tool for investigating TAMs. In M_2_-MACs, the pharmacological inhibition of LRRC8A led to hyperpolarizing responses after a transient depolarization phase, followed by a slight elevation in the intracellular concentration of Ca^2+^. Both the small interfering RNA-mediated and pharmacological inhibition of LRRC8A repressed the transcriptional expression of IL-8 and IL-10, resulting in a significant reduction in their secretion. The inhibition of LRRC8A decreased the nuclear translocation of phosphorylated nuclear factor-erythroid 2-related factor 2 (Nrf2), while the activation of Nrf2 reversed the LRRC8A inhibition-induced transcriptional repression of IL-8 and IL-10 in M_2_-MACs. We identified the CCAAT/enhancer-binding protein isoform B, CEBPB, as a downstream target of Nrf2 signaling in M_2_-MACs. Moreover, among several upstream candidates, the inhibition of nicotinamide adenine dinucleotide phosphate (NADPH) oxidase 2 (NOX2) suppressed the Nrf2–CEBPB transcriptional axis in M_2_-MACs. Collectively, the present results indicate that the inhibition of LRRC8A repressed IL-8 and IL-10 transcription in M_2_-MACs through the NOX2–Nrf2–CEBPB axis and suggest that LRRC8A inhibitors suppress the IL-10-mediated evasion of tumor immune surveillance and IL-8-mediated metastasis and neovascularization in TAMs.

## 1. Introduction

Leucine-rich repeat-containing 8A (LRRC8A) is a component of volume-regulated anion channels (VRACs) and contributes to regulatory volume decreases [1]. VRACs are heteromeric chloride ion (Cl^−^) channels comprising an essential LRRC8A and at least one other LRRC8 member (LRRC8B–E). The intracellular concentration of Cl^−^ varies among cell types and affects the functional outcome of transmembrane Cl^−^ movement through Cl^−^ channels and transporters by regulating the driving force for Cl^−^ conductance [2]. In cells that accumulate Cl^−^ intracellularly, the activation of Cl^−^ channels results in membrane depolarization through Cl^−^ efflux [2,3]. In immune cells, including macrophages, membrane depolarization has been shown to decrease the driving force for Ca^2+^ entry through voltage-independent Ca^2+^ channels, subsequently suppressing intracellular Ca^2+^ signaling [4].

M_2_-polarized, tumor-associated macrophages (TAMs) are mainly derived from monocytes and abundantly infiltrate solid tumors [5]. The localization of TAMs in the tumor microenvironment (TME) correlates with poor clinical outcomes in solid tumors [5]. TAMs represent up to 50% of the tumor mass in solid tumors and support the survival of cancer stem cells by inhibiting immunosurveillance [6]. TAMs, mostly with M_2_-like characteristics in the TME, highly produce the anti-inflammatory cytokine, interleukin 10 (IL-10), cyclooxygenase 2 (COX2), and mannose and scavenger receptors (CD206 and CD163, respectively) and increase the release of ornithine and polyamine through the arginase (Arg) pathway [7,8]. Macrophages differentiated from human monocytic leukemia THP-1 cells are used as human in vitro models to examine M_1_- and M_2_-polarized macrophages [9]. The manipulation of TAM functions is a useful therapeutic approach in clinical practice for the prevention of tumor metastasis [10]. 

IL-10 is highly expressed in TAMs and contributes to the differentiation and recruitment of regulatory T cells (T_reg_s) in tumors [11]. Additionally, IL-8 plays a critical role in the recruitment and accumulation of TAMs and other immunosuppressive cells into the TME, as well as in neovascularization, accelerating tumor proliferation and metastasis in the TME [12]. Elevated numbers of IL-8-expressing TAMs are associated with higher clinical stages and an increased risk of a poor prognosis in cancer patients [13]. We previously demonstrated that the transcription of IL-8 and IL-10 was critically repressed by the activation of the Ca^2+^-activated K^+^ channel, K_Ca_3.1, through the extracellular signal-regulated kinase (ERK) and c-Jun N-terminal kinase (JNK) signaling pathways in M_2_-MACs [14]. 

Nuclear factor-erythroid 2-related factor 2 (Nrf2) is known as a key regulator of oxidative stress, and Nrf2 dissociates from Kelch-like ECH-associated protein 1 (Keap1) and subsequently translocates into the nucleus. The Nrf2-Keap1 signaling pathway plays an important role in the regulation of oxidative stress in cancerous and non-cancerous cells [15,16]. In macrophages, Nrf2 inhibits M_1_ polarization and promotes M_2_ polarization [17,18]. It also upregulates the expression of M_2_ markers such as CD163 and Arg1 [19]. Liu et al. indicated that the inhibition of LRRC8A promoted macrophage phagocytosis by promoting the nuclear translocation of Nrf2 [20]. The Nrf2-mediated regulation of IL-8 and IL-10 expression and secretion remains controversial [21,22,23,24,25]. We previously reported the transcriptional repression of IL-8 and IL-10 by the pharmacological inhibition of Nrf2 in M_2_-MACs [14]; however, the underlying mechanisms have yet to be elucidated.

CCAAT/enhancer-binding protein (C/EBP and CEBP) isoforms are a family of multifunctional basic leucine zipper transcription factors [26], and CEBPB contributes to M_2_ macrophage-specific gene expression, such as IL-10 and Arg1 [27]. Additionally, the overexpression and activation of CEBPs (A, B, and D) stimulate the promoters of the IL-8 and IL-10 genes [28,29,30].

LRRC8A co-localizes with NOX isoforms, such as NOX2 and NOX4, in the plasma membrane and promotes the production of reactive oxygen species (ROS) via these subtypes [31]. NOX2 signaling plays a crucial role in oxidative stress during differentiation into macrophages and in their expression and secretion of pro- and anti-inflammatory mediators [32,33]. Previous studies indicated that Nrf2 was activated in macrophages through the NOX2 signaling pathway [34].

We herein examined the involvement of LRRC8A in the expression and secretion of IL-8 and IL-10 in M_2_-MACs. The results obtained provide novel mechanistic insights into the therapeutic potential of LRRC8A inhibitors in cancer immunotherapy.

## 2. Results

### 2.1. Functional Expression of LRRC8A in M_2_-MACs

As previously reported, high expression levels of the M_2_ marker genes CD163, Arg1, and C–C motif chemokine 22 (CCL22) in M_2_-MACs were confirmed in the present study [14]. We previously demonstrated that the activation of the Ca^2+^-activated K^+^ channel K_Ca_3.1 downregulated the pro-tumorigenic cytokine IL-10 and angiogenic chemokine IL-8/CXCL8 in M_2_-MACs through the ERK-cAMP response element-binding protein (CREB) and JNK-c-Jun cascades [14]. We also reported that the inhibition of Nrf2 downregulated IL-10 and IL-8 in M_2_-MACs; however, the K_Ca_3.1-mediated regulation of these cytokines was not related to the Nrf2 signaling pathway [14]. We herein examined the role of LRRC8A in the regulation of IL-10 and IL-8 in M_2_-MACs. As shown in Figure 1A, LRRC8A transcripts, but not the Ca^2+^-activated Cl^−^ channel (anoctamin 1, ANO1), were highly expressed in M_2_-MACs. Similar results were obtained at the protein level by Western blotting and immunocytochemical imaging (Figure 1B,C). Among the other LRRC8 isoforms, LRRC8B and 8C transcripts were expressed at high levels (Appendix A). Among the Cl^−^ channels/transporters present in intracellular organelles, voltage-dependent anion channel 1 (VDAC1), chloride intracellular channel protein 1 (CLIC1), H^+^/Cl^−^ exchanger transporter 3 (CLCN3), and CLCN7 transcripts were expressed at high levels (Appendix A). The transcriptional expression levels of the isoforms [VDAC2, 3, CLIC2 to 6, K^+^–Cl^−^ co-transporter 1 (KCC1) to 4, cystic fibrosis transmembrane conductance regulator (CFTR), and osteopetrosis-associated transmembrane protein 1 (Ostm1)] were less than 0.01 in arbitrary units (a.u., ratio to β-actin, ACTB) (*n* = 4 for each). Among the Cl^−^ channel modulators expressing at the plasma membrane, chloride channel accessory 1 (CLCA1), CLCA2, CLCA4, and transmembrane protein 206 (TMEM206), the expression levels of CLCA4 and TMEM206 were approximately 0.05 a.u., and those of CLCA1 and CLCA2 were less than 0.01 a.u. (Appendix A). Among the ANO isoforms (ANO1, 2, and 6) that function as Cl^−^ channels, ANO6 transcripts were expressed at high levels (Appendix A).

We then examined the effects of the pharmacological blockade of LRRC8A with endovion (EDV) on membrane potential and [Ca^2+^]_i_. The application of 1 μM EDV induced large membrane hyperpolarization, following slight membrane depolarization (Figure 2A,C,D). Correspondingly, the application of EDV for 10 min (min) induced a significant elevation in [Ca^2+^]_i_ in M_2_-MACs (Figure 2B,E,F). EDV is also known as an ANO1 inhibitor (50% inhibitory concentration (IC_50_) = 0.5 μM for LRRC8A; IC_50_ = 0.7 to 2.0 μM for ANO1). Therefore, we examined the effects of the ANO1-selective inhibitor, ANO1-IN-1, on membrane potential in M_2_-MACs. As shown in Figure 2G,H, no significant responses were observed in M_2_-MACs following the application of 1 μM ANO1-IN-1. The ANO6 inhibitor, abamectin (10 μM), elicited small hyperpolarizing responses in M_2_-MACs (Appendix A).

### 2.2. Decreased Expression and Secretion of IL-8 and IL-10 by the Small Interfering RNA (siRNA)-Mediated and Pharmacological Inhibition of LRRC8A in M_2_-MACs

We examined the effects of the siRNA-mediated inhibition of LRRC8A on the expression and secretion of IL-8 and IL-10 in M_2_-MACs. IL-8 and IL-10 mRNA levels were significantly reduced by the inhibition of LRRC8A (*n* = 4 for each, *p* < 0.01) (Figure 3A,C). Correspondingly, their secretion was significantly decreased by the inhibition of LRRC8A (*n* = 4 for each, *p* < 0.01) (Figure 3B,D). The inhibition efficacy of LRRC8A siRNA was approximately 65% (Appendix A). We also examined the effects of treatment with EDV for 12 h (hr) on their expression and secretion in M_2_-MACs. IL-8 and IL-10 mRNA levels were significantly suppressed by the treatment with 10 μM EDV in a concentration-dependent manner (*n* = 4 for each, *p* < 0.05, 0.01) (Figure 3E,G). Their secretion was also significantly decreased by approximately 50% by treatment with EDV for 24 h (*n* = 4 for each, *p* < 0.01) (Figure 3F,H). No significant changes were observed in their mRNA levels following the pharmacological inhibition of ANO1 and ANO6 (Appendix A) or the siRNA-mediated inhibition of VDAC1, CLIC1, CLCN3, and ANO6 (Appendix A). These results suggest that the inhibition of LRRC8A suppressed the pro-tumorigenic and pro-angiogenic functions of TAMs by repressing the secretion of IL-8 and IL-10.

### 2.3. LRRC8A Inhibition-Induced Transcriptional Repression of IL-8 and IL-10 through the Nrf2 Signaling Pathway in M_2_-MACs

We previously reported that the pharmacological inhibition of Nrf2 with a selective inhibitor, ML385 (5 μM), repressed the transcriptional expression of IL-8 and IL-10 in M_2_-MACs [14]. Similarly, their expression and secretion were significantly suppressed by the siRNA-mediated inhibition of Nrf2 in M_2_-MACs (Figure 4A–D). The inhibitory efficacy of Nrf2 siRNA was approximately 65% (Appendix A). To elucidate whether the LRRC8A inhibition-induced downregulation of IL-8 and IL-10 was mediated through the Nrf2 signaling pathway, we examined the recovery effects of the Nrf2 activator, NK252, on the downregulation of IL-8 and IL-10 in M_2_-MACs. Co-treatment with NK252 (100 μM) and EDV (10 μM) for 12 h reversed the EDV-induced downregulation of these cytokines in M_2_-MACs (*n* = 4, *p* < 0.01 vs. +/−) (Figure 4E,F). A significant increase was observed in IL-8 mRNA levels in M_2_-MACs by a single treatment with NK252 (*n* = 4, *p* < 0.01 vs. −/−); however, no significant change was noted in IL-10 mRNA levels (*n* = 4, *p* > 0.05 vs. −/−) (Figure 4E,F).

To investigate whether the inhibition of LRRC8A reduced the nuclear translocation of P-Nrf2 in M_2_-MACs, the cellular localization of P-Nrf2 was visualized 2 h after the treatment with 10 μM EDV by confocal laser scanning fluorescence microscopy. The anti-P-Nrf2 antibody and nuclei were labeled with the Alexa Fluor 488-conjugated secondary antibody and 4′,6-diamidino-2-phenylindole (DAPI), respectively (Figure 5A). Images from more than 30 cells for each case (*n* = 1) were obtained, and the summarized results are expressed as the percentage of cells with P-Nrf2-positive nuclei. The percentage of these cells was significantly reduced in M_2_-MACs treated with EDV (*n* = 6, *p* < 0.01) (Figure 5A(a,b),B). As a positive control, they were significantly increased by the treatment with NK252 for 2 h (*n* = 6, *p* < 0.01) (Figure 5A(c),C). Nrf2 staining was detected at similar levels in the cytosolic regions of vehicle- (Figure 5D(a)), EDV- (Figure 5D(b)), and NK252-treated (Figure 5D(c)) M_2_-MACs. In M_2_-MACs stained with the Alexa Fluor 488-conjugated anti-rabbit IgG secondary antibody alone, very weak fluorescent signals were observed with the same acquisition settings (Figure 5E). As predicted from our previous findings [14], a co-treatment with NK252 and the selective K_Ca_3.1 activator SKA-121 (10 μM) for 12 h did not reverse the SKA121-induced downregulation of IL-8 and IL-10 in M_2_-MACs (*n* = 4, *p* > 0.05 vs. +/−) (Appendix A), and no significant changes were observed in the percentage of cells with P-Nrf2-positive nuclei (Appendix A) or Nrf2 staining patterns (Appendix A) following the SKA-121 treatment for 2 h (*n* = 6, *p* > 0.05).

### 2.4. Involvement of CEBPB in the LRRC8A Inhibition-Induced Transcriptional Repression of IL-8 and IL-10 in M_2_-MACs

CEBPs are downstream transcriptional modulators of Nrf2, and their activation stimulates the promoter of the IL-8 and IL-10 genes [28,29,30]. Among the six CEBP isoforms, CEBPB was predominantly expressed in M_2_-MACs (*n* = 4) (Figure 6A). As shown in Figure 6B,C, the expression levels of IL-8 and IL-10 were significantly decreased by the siRNA-mediated inhibition of CEBPB in M_2_-MACs (*n* = 4, *p* < 0.01). The inhibition efficacy of CEBPB siRNA was approximately 50% (Appendix A). Furthermore, the expression levels of IL-8 and IL-10 were significantly decreased by the pharmacological inhibition of LRRC8A and Nrf2 with 10 μM EDV and 5 μM ML385, respectively (*n* = 4, *p* < 0.01) (Figure 6D,E). These results indicate that the LRRC8A inhibition-induced transcriptional repression of IL-8 and IL-10 in M_2_-MACs was mediated through the Nrf2–CEBPB transcriptional axis.

### 2.5. Involvement of the NOX2-ROS Axis in the LRRC8A Inhibition-Induced Transcriptional Repression of IL-8 and IL-10 in M_2_-MACs

Previous studies reported that AKT and AMP-activated protein kinase (AMPK) activated the Nrf2 downstream signaling pathway in stimulated RAW264.7 macrophages [35,36]. In the present study, the level of Nrf2 phosphorylation was high in M_2_-MACs (Figure 7A, left panels); however, the level of AKT phosphorylation was very low (Figure 7A, right panels). Correspondingly, no significant changes were observed in the expression levels of IL-8 and IL-10 transcripts in M_2_-MACs following a treatment with the AKT inhibitor AZD5363 (2 μM) for 12 h (Figure 7B,C). Additionally, a significant decrease was not detected in the level of AMPK phosphorylation following LRRC8A inhibition with 10 μM EDV for 2 h (Figure 7D,E). AMPK is activated in macrophages in a Ca^2+^-dependent manner [37]; however, [Ca^2+^]_i_ only slightly increased in M_2_-MACs treated with EDV (Figure 2B). The pharmacological inhibition of AMPK by treatment with Bay3827 (BAY, 10 μM) for 12 h suppressed the expression and secretion of IL-8 and IL-10 and expression of CEBPB in M_2_-MACs (Appendix A). These results suggest that the inhibition of LRRC8A inactivated the Nrf2 downstream signaling pathway through a pathway independent of AMPK. Corresponding to the results in Figure 7D,E, P-AMPK and AMPK were both detected at similar levels in the cytosolic regions of vehicle- (Appendix A and EDV- (Appendix A treated M_2_-MACs. In M_2_-MACs stained with the Alexa Fluor 488-conjugated anti-mouse IgG antibody alone, very weak fluorescent signals were observed under the same acquisition settings (Appendix A).

Apart from AKT and AMPK, ROS also activates the Nrf2 downstream signaling pathway in macrophages [38]. LRRC8A has been shown to activate NOX-dependent ROS production and physically binds to NOX1, 2, and 4 [39,40]. As shown in Figure 8A, among the seven NOX isoforms, NOX2 transcripts were predominantly expressed in M_2_-MACs. NOX2 protein expression was observed in M_2_-MACs (Figure 8B). Correspondingly, the fluorescence signals of the Alexa 488-labeled anti-NOX2 antibody were observed along the plasma membrane of M_2_-MACs (Appendix A). With the same acquisition settings, very weak signals were observed in native THP-1 cells stained with the Alexa 488-labeled anti-NOX2 antibody and in native THP-1 cells and M_2_-MACs stained with the Alexa 488-labeled anti-NOX4 antibody (Appendix A). GLX351322 (GLX) is a NOX4 inhibitor with IC_50_ = 5 μM and also inhibits NOX2 with IC_50_ = 40 μM [40]. On the other hand, GSK2796039 (GSK) is a selective NOX2 inhibitor with IC_50_ = 1 μM [41]. No significant decrease in the expression of IL-8 and IL-10 transcripts was found following the inhibition of NOX4 in M_2_-MACs with 10 μM GLX for 12 h (Figure 8C,D); however, it was significantly decreased by approximately 50% following the inhibition of NOX2 with 100 μM GLX and 10 μM GSK (Figure 8C,D,H,I) for 12 h. Correspondingly, IL-8 and IL-10 secretion levels were reduced following the inhibition of NOX2 (Figure 8F,G,K,L). The expression levels of CEBPB transcripts were also significantly decreased by the inhibition of NOX2 (Figure 8E,J).

Moreover, the cellular localization of P-Nrf2 was visualized 2 h after the inhibition of NOX2 with 100 μM GLX and 10 μM GSK. As shown in Figure 9A,B,D,E, the percentages of cells with P-Nrf2-positive nuclei in M_2_-MACs were significantly reduced by the GLX and GSK treatments (*n* = 6 for each, *p* < 0.01). Nrf2 staining was detected at similar levels in the cytosolic regions of vehicle-, GLX-, and GSK-treated M_2_-MACs (Figure 9C,F).

### 2.6. Decreased ROS Production by LRRC8A Inhibition in M_2_-MACs

We examined the effects of the inhibition of LRRC8A with EDV on intracellular ROS levels in M_2_-MACs using an ROS assay kit. Fluorescence images of the fluorescent compound 2,7-dichlorofluorescein oxidized by ROS were assessed. As shown in Figure 10A,B, fluorescence intensity was significantly decreased by the treatments with 10 μM EDV and 10 μM GSK for 30 min (*n* = 4 for each, *p* < 0.01). Therefore, the inhibition of LRRC8A may downregulate the expression of IL-8 and IL-10 in M_2_-MACs through the NOX2–Nrf2–CEBPB transcriptional axis.

### 2.7. Suppression of the High Extracellular K^+^-Induced Upregulation of IL-8 and IL-10 by LRRC8A and NOX2 Inhibition in M_2_-MACs

In a hypoxic TME, the death of cancer cells leads to an increase in K^+^ in the extracellular compartment, and high intratumoral K^+^ promotes immunosuppressive potency [42,43]. Additionally, we previously reported the high extracellular K^+^ ([K^+^]_e_)-induced upregulation of IL-8 and IL-10 in M_2_-MACs [14]. The concentration of K^+^ in Roswell Park Memorial Institute (RPMI)-1640 medium was 5 mM, and media with high K^+^ (final concentration: 35 mM) were prepared by supplementation with 30 mM KCl. As previously reported [14], real-time PCR and ELISA experiments showed increases in the transcriptional expression (*n* = 4, *p* < 0.01) (Figure 11A,B,F,G) and secretion (*n* = 4, *p* < 0.01) (Figure 11D,E,I,J) of IL-10 and IL-8 in M_2_-MACs exposed to high [K^+^]_e_. Correspondingly, the expression levels of CEBPB were increased by high [K^+^]_e_ (Figure 11C,H). No significant changes in their expression and secretion were noted by supplementation with 30 mM NaCl instead of 30 mM KCl (*n* = 4, *p* > 0.05) (Appendix A). High [K^+^]_e_-induced increases in IL-8 and IL-10 expression and secretion were significantly reduced by simultaneous treatments with 10 μM EDV (*n* = 4, *p* < 0.01) (Figure 11A,B,D,E) and 10 μM GSK (*n* = 4, *p* < 0.01) (Figure 11F,G,I,J). Correspondingly, high [K^+^]_e_-induced increases in CEBPB expression were reduced by treatments with EDV (Figure 11C) and GSK (Figure 11H). These results demonstrated that the inhibition of LRRC8A in TAMs may suppress overexpressed IL-10 and IL-8 in a high [K^+^]_e_-exposed TME.

### 2.8. Mechanisms Underlying the Transcriptional Upregulation of LRRC8A in M_2_-MACs

Chen et al. recently reported that LRRC8A was upregulated through mRNA modification by NOP2/Sun RNA methyltransferase 2 (NSUN2), an m^5^C methyltransferase, and the stability of the m^5^C-modified LRRC8A mRNA was further enhanced by binding to Y-box binding protein 1 (YBX1) [44]. As shown in Figure 12A, LRRC8A was upregulated by M_2_ differentiation in THP-1 cells. In addition, NSUN2 and YBX1 were highly expressed in M_2_-MACs, similar to native THP-1 and M_0_-MACs (Figure 12B,C). However, the siRNA-mediated inhibition of NSUN2 and pharmacological inhibition of YBX1 with SU056 (10 μM) did not induce LRRC8A mRNA destabilization in M_2_-MACs (Figure 12D,E). On the other hand, histone deacetylases (HDACs) have been shown to play crucial roles in macrophage functions [45] and are responsible for the post-transcriptional modification of ion channels such as ANO1 and K_Ca_3.1 [46,47]. As shown in Figure 12F,G, HDAC1, 2, and 3 and sirtuin 1 (SIRT1) transcripts were expressed in M_2_-MACs at high levels. Among them, HDAC3 levels were approximately four-fold higher in M_2_-MACs than in native THP-1 and M_0_-MACs (Figure 12F,G and Appendix A). The expression levels of LRRC8A transcripts were reduced by approximately 50% following treatments with the pan-HDAC inhibitor, vorinostat (1 μM), and the selective HDAC3 inhibitor, T247 (1 μM), for 24 h; however, no significant changes were observed following treatments with the HADC1/2 inhibitor, AATB (10 μM), and the SIRT1 inhibitor, Ex527 (1 μM) (Figure 12H). Correspondingly, the siRNA-mediated inhibition of HDAC3 significantly reduced their levels (Figure 12I). These results suggest that HDAC3 is, at least in part, responsible for the upregulated expression of LRRC8A in M_2_-MACs.

## 3. Discussion

TAMs are a major component of the TME and maintain an immunosuppressive microenvironment by secreting pro-tumorigenic and pro-angiogenic factors, including IL-10 and IL-8 [11,12,13]. We previously reported that the activation of the Ca^2+^-activated K^+^ channel K_Ca_3.1 repressed the transcriptional expression of IL-10 and IL-8 through the JNK and ERK signaling pathways in M_2_-MACs [14]. However, the mechanisms underlying the transcriptional regulation of IL-10 and IL-8 through the Nrf2 signaling pathway in M_2_-MACs remain unclear. Consistent with our previous findings [14], the pharmacological activation of K_Ca_3.1 with SKA-121 did not suppress the nuclear translocation of P-Nrf2, and Nrf2 activation with NK252 did not reverse the K_Ca_3.1 activation-induced transcriptional repression of IL-8 and IL-10 in M_2_-MACs (Appendix A). LRRC8A is associated with phenotype polarization and phagocytosis in macrophages [18] and is physically coupled with NOX isoforms [31,39,48]. The NOX–ROS–Nrf2 axis is important for M_2_ macrophage activation [49]. The present study elucidated the critical role of LRRC8A in the Nrf2-mediated production of IL-8 and IL-10 in M_2_-MACs. The main results obtained in the present study are as follows. (1) LRRC8A expression was higher in M_2_-MACs than in native THP-1 and M_0_-MACs. The pharmacological inhibition of LRRC8A induced prolonged hyperpolarization and a sustained elevation in basal [Ca^2+^]_i_ in M_2_-MACs (Figure 1, Figure 2 and Figure 12). The overexpression of HDAC3 may be associated with the upregulation of IL-8 and IL-10 in M_2_-MACs (Figure 12). (2) The inhibition of LRRC8A repressed the expression and secretion of IL-8 and IL-10 through the Nrf2–CEBPB transcriptional axis in M_2_-MACs (Figure 3, Figure 4, Figure 5 and Figure 6). (3) The inhibition of LRRC8A suppressed the activation of Nrf2 through the NOX2-ROS signaling pathway in M_2_-MACs (Figure 8, Figure 9 and Figure 10). (4) The upregulation of IL-8 and IL-10 following the exposure of M_2_-MACs to high K^+^ mimicking the TME was mostly reversed by the inhibition of LRRC8A (Figure 11). Cytosolic Cl^−^ is increased by the inhibition of LRRC8A through a reduction in Cl^−^ efflux. As shown in Figure 2A,D, the LRRC8A inhibition-induced accumulation of [Cl^−^]_i_ contributed to plasma membrane hyperpolarization in M_2_-MACs. In myoblasts expressing voltage-gated Ca^2+^ channels, the inhibition of LRRC8A decreased basal [Ca^2+^]_i_ [50]; however, in M_2_-MACs, LRRC8A inhibition-induced hyperpolarization elevated [Ca^2+^]_i_ through voltage-independent Ca^2+^ channels such as Orai/STIM and transient receptor potential (TRP) channels [51,52].

The activation of the PI3K-AKT signaling pathway is associated with M_2_ macrophage polarization and activates Nrf2 downstream in macrophages [53,54]. The inhibition of LRRC8A mitigated the activation/phosphorylation of AKT in T cells [17]. Therefore, we predicted the possible contribution of AKT to the LRRC8A inhibition-induced inactivation of Nrf2 in M_2_-MACs. However, the expression level of P-AKT was very low in M_2_-MACs (Figure 7A), and no significant changes in the expression of IL-8 and IL-10 were found by AKT inhibition in M_2_-MACs (Figure 7B,C). Liu et al. showed that the inhibition of LRRC8A promoted macrophage phagocytosis by activating AMPK, resulting in the nuclear translocation of Nrf2 [55]. However, AMPK is not distinguished as a Cl^−^-sensitive and volume-sensitive protein kinase. AMPK is activated by [Ca^2+^]_i_ elevation through the activation of Ca^2+^/calmodulin-dependent protein kinase kinase 2 (CaMKK2) [55]. In the present study, LRRC8A inhibition-induced hyperpolarization slightly increased intracellular steady-state [Ca^2+^]_i_ in M_2_-MACs (Figure 2); however, the phosphorylation levels of AMPK were not changed by the inhibition of LRRC8A with EDV (Figure 7 and Appendix A).

WNK lysine-deficient protein kinase 1 (WNK1) is a Cl^−^-sensitive protein kinase known as an upstream molecule of AMPK. The activation of WNK1 inhibits the AMPK signaling pathway [56,57]. In M_2_-MACs, among four WNK isoforms (WNK1–4), WNK1 was highly expressed; however, no significant changes in the expression levels of IL-8 and IL-10 were found by the pharmacological and siRNA-mediated inhibition of WNK1 in M_2_-MACs (Appendix A). The expression levels of the other isoforms (WNK2–4) were less than 0.0001 a.u. in M_2_-MACs. As described above, the inhibition of LRRC8A did not affect the phosphorylation level of AMPK in M_2_-MACs. However, previous studies indicated that the AMPK–Nrf2 axis is an important upstream signaling pathway for cytokine expression such as IL-10 [58,59]. As shown in Appendix A, the pharmacological inhibition of AMPK suppressed IL-8 and IL-10 transcription through the Nrf2–CEBPB transcriptional axis. These results suggest that AMPK inhibitors suppress these cytokines through mechanisms independent of LRRC8A. In the present study, we focused on the NOX–ROS–Nrf2 transcriptional axis, but not the WNK1–AMPK–Nrf2 one, as the downstream of LRRC8A in M_2_-MACs.

LRRC8A is physically associated with NOX1, 2, and 4, and is required for NOX activity [31,39,48]. Macrophage-specific NOX2-derived ROS play an essential role in Nrf2 activation and M_2_ polarization [49]. In cancer cells, Nrf2 suppresses tumor promotion or conversely exerts pro-tumorigenic effects [60]; however, the Nrf2 signaling pathway in M_2_-MACs prevents cancer immunosurveillance by producing pro-tumorigenic cytokines. In the present study, NOX2 expression was higher in M_2_-MACs than NOX1 and NOX4 expression and was distributed along the plasma membrane (Figure 8A,B and Appendix A). The expression and secretion levels of IL-8 and IL-10 and the nuclear translocation of P-Nrf2 were significantly reduced by the inhibition of NOX2 (Figure 8 and Figure 9). Additionally, ROS levels were reduced by the inhibition of LRRC8A and NOX2 (Figure 10).

CEBPB contributes to M_2_ macrophage polarization by promoting the expression of M_2_ marker genes such as Arg1 and immunosuppressive enzymes including NOX2 [5,27,61]. In the present study, CEBPB was predominantly expressed in M_2_-MACs (Figure 6A), and the Nrf2–CEBPB transcriptional axis was involved in the LRRC8A inhibition-mediated downregulation of IL-8 and IL-10 (Figure 6B–E). The present study is the first to provide evidence for the LRRC8A-mediated regulation of the Nrf2–CEBPB transcriptional axis. Therefore, the LRRC8A–NOX2–Nrf2–CEBPB transcriptional axis may play a key role in M_2_ polarization by regulating the gene expression of M_2_-related molecules. As shown in Appendix A, the pharmacological and/or siRNA-mediated inhibition of LRRC8A, Nrf2, and CEBPB in M_2_-MACs significantly reduced the expression levels of NOX2, CD169, and Arg1.

ANO6, VDAC1, and CLIC1 transcripts were highly expressed in M_2_-MACs (Appendix A). However, their pharmacological and/or siRNA-mediated inhibition did not affect the expression levels of IL-8 and IL-10 in M_2_-MACs (Appendix A). The involvement of ANO6 and VDAC1 in the Nrf2 signaling pathway has not yet been demonstrated. In vascular endothelial cells, CLIC1 inhibits the nuclear translocation and phosphorylation of Nrf2 [62,63]. In macrophages, CLIC1 plays an essential role in phagosome acidification and superoxide production [64]; however, it remains unclear whether CLIC1 regulates the upstream signaling pathways of IL-8 and IL-10 transcription.

A recent study reported that LRRC8A was upregulated by NSUN2-mediated m^5^C modification in cancer cells, and m^5^C-modified LRRC8A mRNA increased RNA stability by binding to YBX1 [44]. However, neither siRNA-mediated NSUN2 inhibition nor YBX1 inhibition with SU056 downregulated LRRC8A (Figure 12D,E). A previous study reported that the expression of NSUN2 negatively correlated with that of M_2_ macrophage markers, such as CD163, and its methylation inhibited the polarization of M_2_ macrophages [65]. Therefore, it is reasonable to assume that NSUN2 is not related to the transcriptional regulation of LRRC8A in M_2_-MACs. On the other hand, HDAC3 was shown to play a significant role in macrophage development and homeostasis [45,66]. We found that the gene expression of K_Ca_3.1 and ANO1 was epigenetically regulated by HDAC3 in breast and prostate cancers [46,47]. As shown in Figure 12F, HDAC3 was upregulated in M_2_-MACs, and the selective inhibition of HDAC3, but not HDAC1/2 or SIRT1, significantly reduced the expression level of LRRC8A transcripts in M_2_-MACs (Figure 12H,I). The present study is the first to provide evidence for the HDAC3-mediated epigenetic modification of LRRC8A.

## 4. Materials and Methods

### 4.1. Chemicals and Reagents

The human acute monocytic leukemia cell line, THP-1, was supplied by the Cell Resource Center for Biomedical Research, Tohoku University (Sendai, Japan). RPMI-1640 medium (189-02025) and 0.25w/v% trypsin-1mmol/L EDTA·4Na solution (201-16945) were purchased from FUJIFILM Wako Pure Chemicals (Osaka, Japan). Fetal bovine serum, phorbol 12-myristate 13-acetate (PMA) (Product code: P1585), GLX351322 (SML2546), abamectin (31732), and DAPI (D2542) were from Sigma-Aldrich (St. Louis, MO, USA). Recombinant human IL-4 (AF-200-04) and IL-13 (AF-200-13) were from PeproTech (Cranbury, NJ, USA). DiBAC_4_(3) (D545), Fura 2-AM (F015), and the ROS assay kit (photo-oxidation-resistant DCFH-DA) (R253) were from Dojindo (Kumamoto, Japan). The Luna Universal qPCR Master Mix (M3003E) was from New England Biolabs Japan (Tokyo, Japan). PCR primers were from Nihon Gene Research Laboratories (Miyagi, Japan). The ECL Advance Chemiluminescence Kit (02230-30) was from Nacalai Tesque (Kyoto, Japan). EDV (HY-105917), ANO1-IN-1 (HY-16320), NK252 (HY-19734), SU056 (HY-50231), and SKA121 (HY-107414) were from MedChemExpress (Monmouth Junction, NJ, USA). IL-10/IL-8 human uncoated enzyme-linked immunosorbent assay (ELISA) kits (88-7106-77, 88-8086-77), the Halt^TM^ phosphatase inhibitor cocktail (SF-78420), and validated/pre-designed siRNAs for LRRC8A (siRNA ID: s32107, pre-designed), Nrf2 (s9491, pre-designed), VDAC1 (s14768, validated), CLIC1 (s636, validated), ANO6 (s225825, pre-designed), CLCN3 (s3136, validated), CEBPB (s2892, pre-designed), NSUN2 (s29683, pre-designed), and WNK1 (s35234, validated) were from Thermo Fisher Scientific (Waltham, MA USA). siRNAs for control-A (sc-37007) and HDAC3 (sc-35538) were from Santa Cruz Biotechnology, Sant Cruz, CA, USA. AZD5363 (15406), WNK-IN-11 (29676), and GSK2795039 (33777) were from Cayman Chemical (Ann Arbor, MI, USA). ML385 (S8790), BAY3827 (S9833), and Ex527 (S1541) were from Selleckchem (Yokohama, Japan). The CytoFix/Perm Kit was from BD Pharmingen (AB_2869008) (Franklin Lakes, NJ, USA). PCR primers were from Nihon Gene Research Laboratories (Sendai, Japan). The HDAC inhibitors were supplied by Professor Suzuki (Osaka University, Japan). Other chemicals and reagents were from Sigma-Aldrich, FUJIFILM Wako Pure Chemicals, and Nacalai Tesque.

### 4.2. Cell Culture and Differentiation into M_2_ Macrophages

The differentiation of the human monocytic leukemia cell line, THP-1, into M_0_ macrophages was induced by treatment with PMA (100 ng/mL) for 8 h. After removal of the medium, cells were incubated with RPMI-1640 medium supplemented with IL-4 and IL-13 (20 ng/mL each) for 72 h to induce the polarization of M_2_ macrophages [14]. Drug applications were performed 48 (for Western blotting) or 60 (for qPCR) hr after the incubation with IL-4/IL-13.

### 4.3. Measurements of Membrane Potential and Intracellular Concentration of Ca^2+^ ([Ca^2+^]_i_)

Membrane potential was measured using the fluorescent voltage-sensitive dye, DiBAC_4_(3) [14]. In membrane potential imaging, cells loaded with DiBAC_4_(3) were illuminated at a wavelength of 490 nm. [Ca^2+^]_i_ was measured using the fluorescent Ca^2+^ indicator dye, Fura 2-AM, and cells loaded with Fura 2 were alternatively illuminated at wavelengths of 340 and 380 nm. The fluorescent intensity of Fura 2 was expressed as measured 340/380 nm fluorescence ratios after background subtraction. After the adherent cells were treated with trypsin solution, isolated cells were seeded onto the glass bottom dish and incubated with the medium for 4–6 h. Before fluorescence measurements, cells were incubated in extracellular solution containing both 100 nM DiBAC_4_(3) and 10 μM Fura 2-AM for 30 min. The extracellular solution was composed of the following (in mM): 137 NaCl, 5.9 KCl, 2.2 CaCl_2_, 1.2 MgCl_2_, 14 glucose, and 10 4-(2-hydroxyethyl)piperazine-1-ethanesulfonic acid (pH 7.4). Fluorescence images were recorded using the ORCA-Flash2.8 digital camera (Hamamatsu Photonics, Hamamatsu, Japan). Data collection and analyses were performed using an HCImage system (Hamamatsu Photonics). Images were obtained every 5 s, and fluorescent intensity values were assessed using the average for 1 min (12 images). Drug solutions were applied by operating the three-way stopcock.

### 4.4. Real-Time PCR Assay

Total RNA extraction and cDNA synthesis were conducted as previously reported [14]. Gene-specific primers for real-time PCR examinations were designed using Primer Express^TM^ software (Ver 1.5, Thermo Fisher Scientific). Real-time PCR was performed using Luna Universal qPCR Master Mix (New England Biolabs Japan, Tokyo, Japan) on an ABI 7500 real-time PCR instrument (Thermo Fisher Scientific) [14]. PCR primers of human origin are listed in Appendix A. Unknown quantities relative to the standard curve for a particular set of primers were calculated [14], yielding the transcriptional quantitation of gene products relative to the ACTB.

### 4.5. Western Blots

Whole-cell lysates of M_2_-MACs were extracted by RIPA buffer with phosphatase inhibitor cocktail at a final concentration of 1%. Equal amounts of protein were subjected to SDS-PAGE and immunoblotting with the antibodies shown in Appendix A and were then incubated with an anti-rabbit or anti-mouse HRP-conjugated IgG secondary antibody. The ECL Advance Chemiluminescence Reagent Kit was used to identify the bound antibody. The resulting images were analyzed using an Amersham Imager 600 (GE Healthcare Japan) [14]. The optical density of the protein band signal relative to that of the ACTB signal was calculated using ImageJ software (Ver. 1.42, NIH, USA), and protein expression levels in the vehicle control are expressed as 1.0.

### 4.6. Measurement of Cytokine Production by ELISA and ROS by a ROS Assay Kit

Human IL-10 and IL-8 levels in supernatant samples were measured with the respective IL-10/IL-8 Human Uncoated ELISA Kits, according to the manufacturer’s protocols. Standard curves were plotted using a series of cytokine/chemokine concentrations [14]. Absorbance was measured using the microplate reader SpectraMax 384 (Molecular Devices Japan, Tokyo, Japan) at a test wavelength of 450 nm and a reference wavelength of 650 nm. To measure intracellular ROS levels, isolated cells with trypsin solution were seeded onto the glass bottom dish for 4–6 h and then were loaded with 10 μM photo-oxidation-resistant DCFH-DA dye at 37 °C for 30 min, according to the manufacturer’s instruction. The dyes were washed out by the extracellular solution (see above) and illuminated at a wavelength of 490 nm. Fluorescence images of the fluorescent compound 2,7-dichlorofluorescein oxidized by ROS were recorded using the ORCA-Flash2.8 digital camera (Hamamatsu Photonics). Data collection and analyses were performed using the HCImage system (Hamamatsu Photonics).

### 4.7. Confocal Imaging of the Nuclear and Cytosolic Distribution of Nrf2 and AMPK and the Plasma Membrane Localization of LRRC8A and NOX2

After the treatment with trypsin solution, isolated M_2_-MACs were fixed and permeabilized using a CytoFix/Perm Kit. The antibodies shown in Appendix A were labeled with an Alexa Fluor 488-conjugated secondary antibody. After seeding onto a glass bottom dish, fluorescence images were visualized using a confocal laser scanning microscopy system (Nikon A1R, Tokyo, Japan) [14]. Image data were quantitatively analyzed using ImageJ software. To assess the P-Nrf2-positive cells, the images from more than 30 cells for each case (*n* = 1) were obtained.

### 4.8. Statistical Analysis

Statistical analyses were performed with the statistical software XLSTAT (version 2013.1). To assess the significance of differences between two groups and among multiple groups, the unpaired/paired Student’s *t*-test with Welch’s correction or a one-way ANOVA with Tukey’s test was used. Results with a *p* value < 0.05 were considered to be significant. Data are shown as means ± standard error.

## 5. Conclusions

[K^+^]_e_ elevation by necrotic cancer and cancer-infiltrating cells in the TME increased the expression and production of IL-10 and IL-8 in M_2_-MACs. In addition to our previous findings showing the effectiveness of the activation of K_Ca_3.1 [14], the present study demonstrated the effectiveness of the inhibition of LRRC8A for TME-mediated TAM dysregulation, which induced the accumulation of pro-tumorigenic and pro-angiogenic cytokines, such as IL-8 and IL-10. However, the clinical and therapeutic importance of LRRC8A inhibitors remains unclear. The present study provides novel insights that support the potential of potent and selective LRRC8A inhibitors as therapeutic applications in TAM-targeting cancer immunotherapy. Recent studies indicated that TAM-like M_2_ macrophages enhanced resistance to anti-cancer drugs and the recruitment of immunosuppressive cells, such as T_reg_s and myeloid-derived suppressor cells (MDSCs), to the TME, and also that these effects were associated with LRRC8A [67,68]. Therefore, the inhibition of LRRC8A may overcome resistance to anti-cancer drugs by downregulating drug-metabolizing enzymes and efflux transporters and may also prevent the accumulation of immunosuppressive cells (TAMs, T_reg_s, and MDSCs) by suppressing the production of their recruiting chemokines. Further studies under conditions that mimic the TME are needed to examine the clinical applications and limitations of LRRC8A inhibitors in cancer immunotherapy.

## Figures and Tables

**Figure 1 ijms-25-09612-f001:**
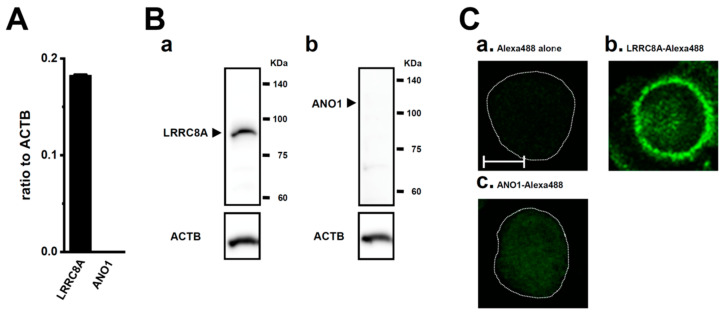
Expression of LRRC8A in M_2_-MACs. (**A**) Real-time PCR examination of LRRC8A and ANO1 in M_2_-MACs. Expression levels are shown as the ratio to the ACTB (*n* = 4 for each). (**B**) Protein expression of LRRC8A (a) and ANO1 (b) in M_2_-MACs protein lysates. Blots were probed with anti-LRRC8A (approx. 95 kDa), anti-ANO1 (approx. 115 kDa), and anti-ACTB (approx. 45 kDa) antibodies. (**C**) Confocal fluorescence images (green) of Alexa Fluor 488-labeled LRRC8A (b) and ANO1 (c) and negative control with Alexa Fluor 488 alone (a) in M_2_-MACs. Dashed lines show cell boundaries. The scale bar in ‘(a)’ shows 10 μm (same in (b,c)).

**Figure 2 ijms-25-09612-f002:**
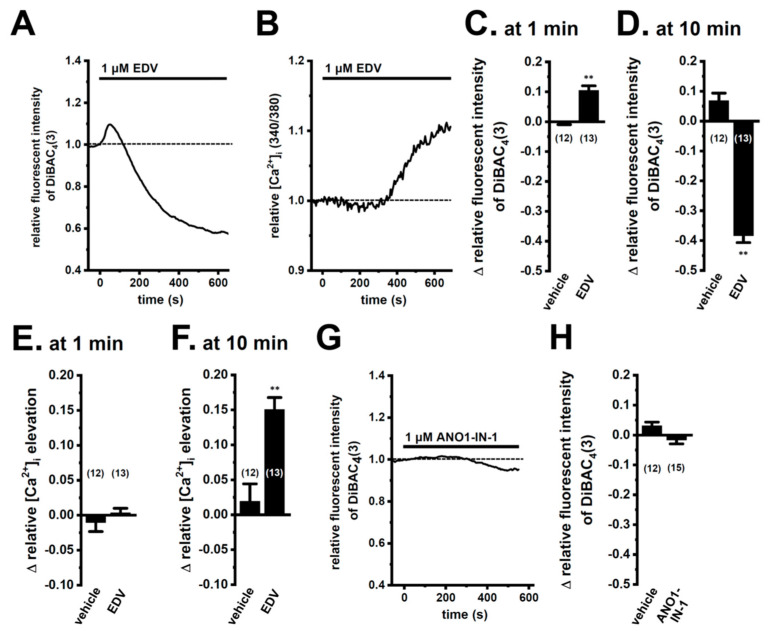
Functional expression of LRRC8A in M_2_-MACs. (**A**,**B**) Simultaneous measurement of changes in membrane potential (**A**) and [Ca^2+^]_i_ (**B**) following the application of the LRRC8/ANO1 inhibitor, endovion (EDV, 1 μM), using bis-(1,3-dibutylbarbituric acid)trimethine oxonol [DiBAC_4_(3)] and Fura 2-acetoxymethyl ester (Fura 2-AM), respectively. The relative time course of changes in fluorescence intensities (1.0 at time 0 s) from an M_2_-MAC is shown. (**C**,**D**) Summarized results of EDV (1 μM)-induced depolarization (at 1 min) (**C**) and hyperpolarization (at 10 min) (**D**) responses. (**E**,**F**) Summarized results of EDV (1 μM)-induced changes in [Ca^2+^]_i_ at 1 min (**E**) and 10 min (**F**). (**G**,**H**) Measurement of changes in membrane potential following the application of the selective ANO1 inhibitor, ANO1-IN-1 (1 μM), using DiBAC_4_(3). The relative time course of changes in fluorescence intensities (1.0 at time 0 s) from an M_2_-MAC is shown (**G**). Summarized results of ANO1-IN-1-induced responses (**H**). Numbers used for experiments are shown in parentheses. **: *p* < 0.01 vs. the vehicle control.

**Figure 3 ijms-25-09612-f003:**
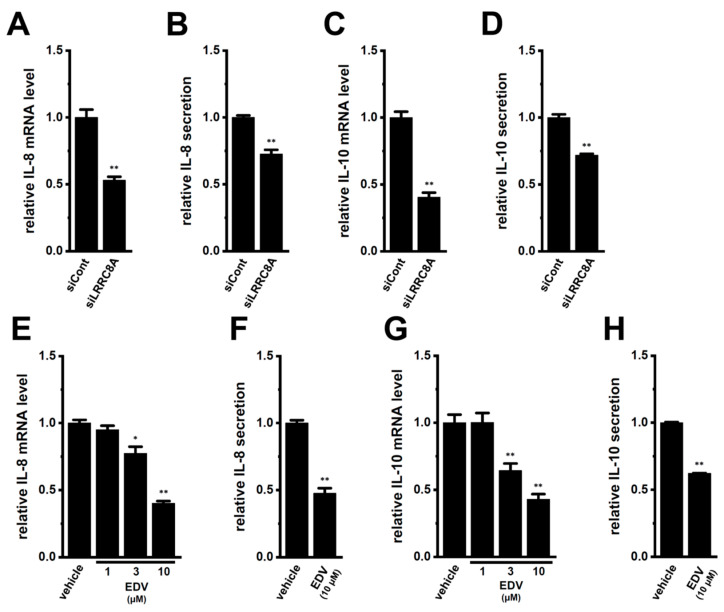
Effects of the siRNA-mediated and pharmacological inhibition of LRRC8A on IL-8 and IL-10 expression and secretion in M_2_-MACs. (**A**,**C**) Real-time PCR examination of IL-8 (**A**) and IL-10 (**C**) expression in M_2_-MACs 48 h after the transfection of LRRC8A siRNA (siLRRC8A). The mRNA expression level in the control siRNA (siCont)-transfected group is expressed as 1.0 (*n* = 4 for each). (**B**,**D**) Quantitative detection of IL-8 (**B**) and IL-10 (**D**) secretion by an ELISA assay in the siCont and siLRRC8A groups. The cytokine secretion level in the siCont group is expressed as 1.0 (*n* = 4 for each). (**E**,**G**) Real-time PCR examination of IL-8 (**E**) and IL-10 (**G**) expression in vehicle- and endovion (EDV: 1, 3, and 10 μM)-treated M_2_-MACs for 12 h. The mRNA expression level in the vehicle control is expressed as 1.0 (*n* = 4 for each). (**F**,**H**) Quantitative detection of IL-8 (**F**) and IL-10 (**H**) secretion by an ELISA assay in the vehicle- and EDV (10 μM)-treated groups. The cytokine secretion level in the vehicle control is expressed as 1.0 (*n* = 4 for each). *, **: *p* < 0.05, 0.01 vs. siCont and the vehicle control.

**Figure 4 ijms-25-09612-f004:**
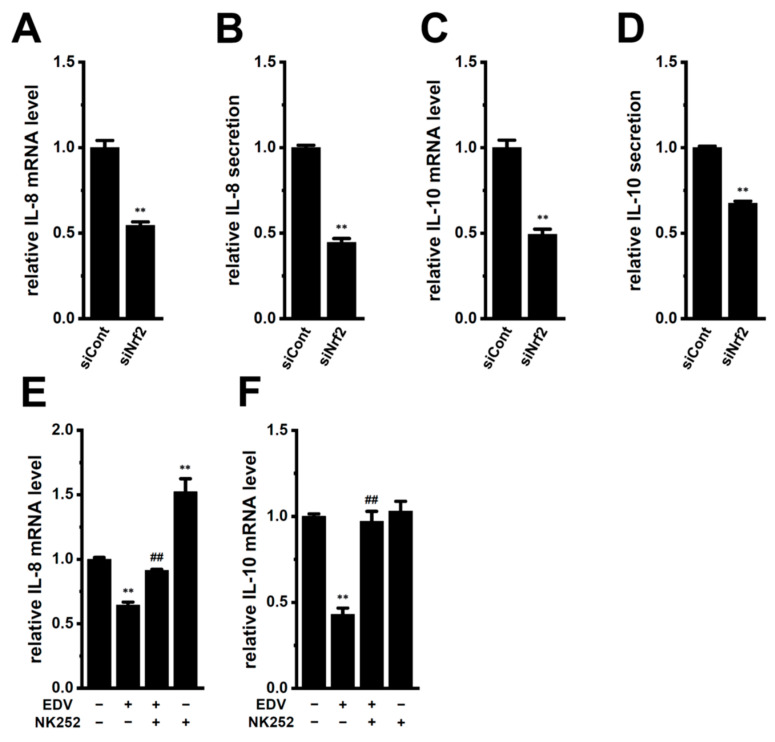
Effects of the siRNA-mediated inhibition of Nrf2 on IL-10 and IL-8 expression and secretion and effects of the activation of Nrf2 on EDV-induced decreases in their expression levels in M_2_-MACs. (**A**,**C**) Real-time PCR examination of IL-8 (**A**) and IL-10 (**C**) expression in M_2_-MACs 48 h after the transfection of Nrf2 siRNA (siNrf2). The mRNA expression level in the control siRNA (siCont)-transfected group is expressed as 1.0 (*n* = 4 for each). (**B**,**D**) Quantitative detection of IL-8 (**B**) and IL-10 (**D**) secretion by an ELISA assay in the siCont- and siNrf2-transfected groups. The cytokine secretion level in the siCont group is expressed as 1.0 (*n* = 4 for each). (**E**,**F**) Real-time PCR examination of IL-8 (**E**) and IL-10 (**F**) in M_2_-MACs treated (+) or untreated (−) with 10 μM endovion (EDV) and 100 μM NK252, the Nrf2 activator for 12 h (*n* = 4 for each). After normalization to ACTB mRNA expression levels, IL-8 and IL-10 mRNA expression levels in the vehicle control (−/−) are expressed as 1.0. **: *p* < 0.01 vs. siCont and −/−; ^##^: *p* < 0.01 vs. +/−.

**Figure 5 ijms-25-09612-f005:**
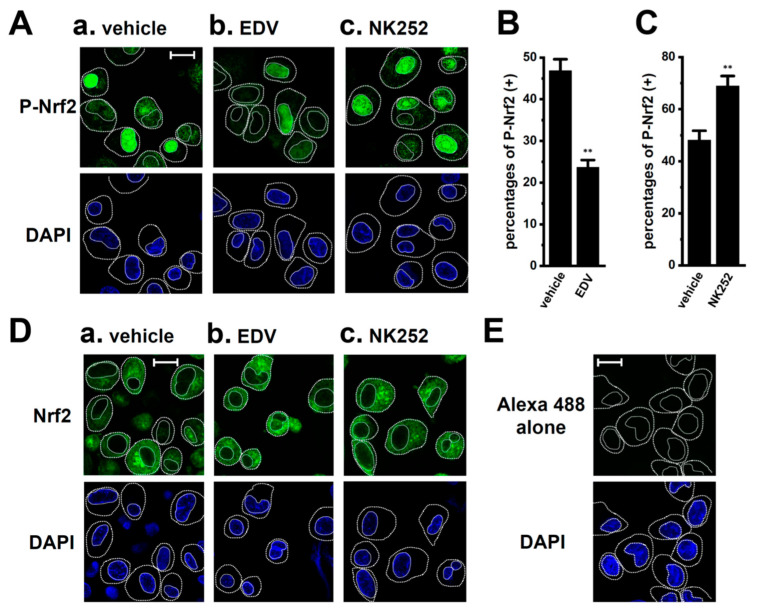
Effects of EDV on the nuclear translocation of P-Nrf2 in M_2_-MACs. (**A**) Confocal fluorescent images of Alexa Fluor 488-labeled P-Nrf2 in vehicle-treated (a), endovion- (EDV) (10 μM) (b), and NK252-treated (100 μM) (c) M_2_-MACs for 2 h (upper panels, single green image). (**B**,**C**) Summarized results of the percentages of P-Nrf2-positive [P-Nrf2(+)] M_2_-MACs in nuclei (*n* = 6 for each). In each batch (*n* = 1), more than 30 cells treated with EDV (**B**) and NK-252 (**C**) were observed by confocal laser scanning microscopy. (**D**) Confocal fluorescent images of Alexa Fluor 488-labeled Nrf2 in vehicle- (a), EDV (b)-, and NK252 (c)-treated M_2_-MACs (upper panels, single green image). (**E**) Confocal fluorescent images of Alexa Fluor 488 alone in vehicle-treated M_2_-MACs. Nuclear morphologies are shown by DAPI images (lower panels). Thick and thin dashed lines show the plasma membrane and nuclear boundary, respectively. The scale bars in ‘**A**(a)’ (same in **A**(b,c)) and ‘**D**(a)’ (same in **D**(b,c)) show 20 μm. **: *p* < 0.01 vs. the vehicle control.

**Figure 6 ijms-25-09612-f006:**
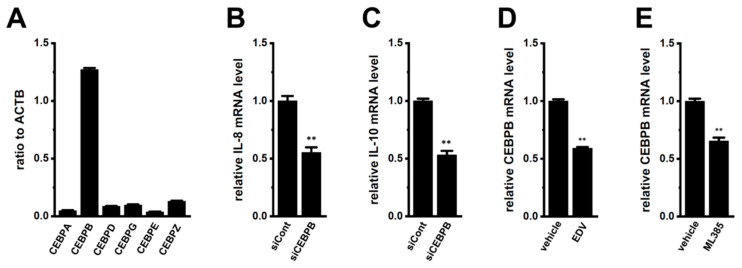
Identification of CEBP isoforms as downstream transcriptional factors of Nrf2 in M_2_-MACs. (**A**) Real-time PCR examination of CEBP isoform expression in M_2_-MACs. Expression levels are shown as the ratio to the ACTB (*n* = 4 for each). (**B**,**C**) Real-time PCR examination of IL-8 (**B**) and IL-10 (**C**) expression in M_2_-MACs 48 h after the transfection of CEBPB siRNA (siCEBPB). The mRNA expression level in the control siRNA (siCont)-transfected group is expressed as 1.0 (*n* = 4 for each). (**D**,**E**) Real-time PCR examination of CEBPB expression in vehicle-, endovion (EDV; 10 μM)- (**D**), and ML385 (5 μM)-treated (**E**) M_2_-MACs for 12 h. The mRNA expression level in the vehicle control is expressed as 1.0 (*n* = 4 for each). **: *p* < 0.01 vs. siCont and the vehicle control.

**Figure 7 ijms-25-09612-f007:**
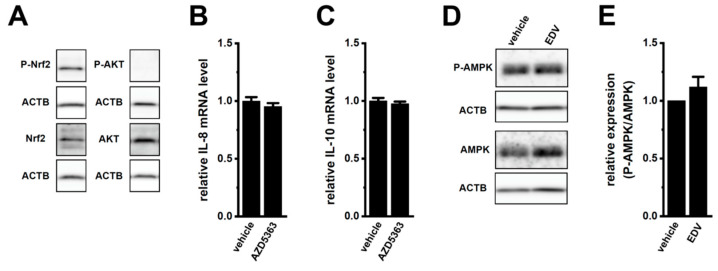
No involvement of AKT or AMPK in the LRRC8A inhibition-induced downregulation of IL-8 and IL-10 in M_2_-MACs. (**A**) Protein expression of P-Nrf2, Nrf2, P-AKT, and AKT in protein lysates of M_2_-MACs. Blots were probed with anti-P-Nrf2/Nrf2 (approx. 100 kDa), anti-P-AKT/AKT (approx. 60 kDa), and anti-ACTB (approx. 45 kDa) antibodies. (**B**,**C**) Real-time PCR examination of IL-8 (**B**) and IL-10 (**C**) expression in vehicle- and AKT inhibitor AZD5363 (2 μM)-treated M_2_-MACs for 12 h. The mRNA level in the vehicle control is expressed as 1.0 (*n* = 4 for each). (**D**) Protein expression of P-AMPK and AMPK in protein lysates of vehicle- and endovion (EDV; 10 μM)-treated M_2_-MACs for 2 h. Blots were probed with anti-P-AMPK/AMPK (approx. 65 kDa) and anti-ACTB (approx. 45 kDa) antibodies. (**E**) Summarized results of the relative expression of P-AMPK/AMPK were obtained from the optical density of P-AMPK, AMPK, and ACTB band signals (*n* = 4 for each).

**Figure 8 ijms-25-09612-f008:**
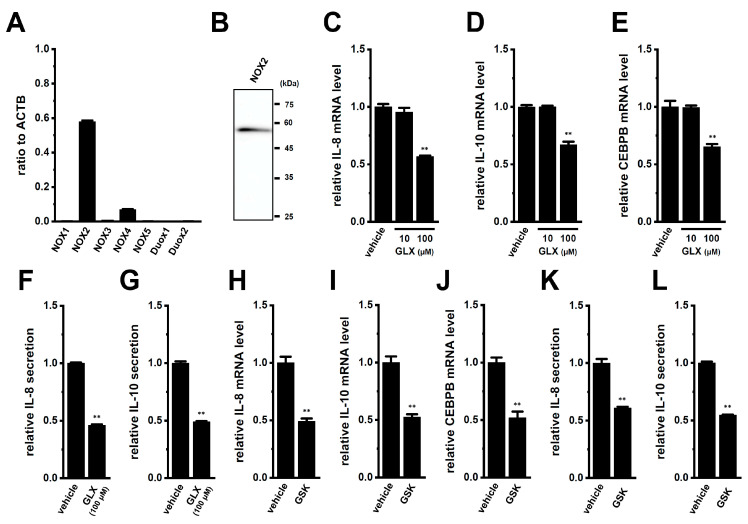
Involvement of NOX2 in the LRRC8A inhibition-induced downregulation of IL-8 and IL-10 in M_2_-MACs. (**A**) Identification of NOX isoforms mainly expressed in M_2_-MACs by a real-time PCR examination. Expression levels are shown as the ratio to the ACTB. The mRNA expression level in the vehicle control is expressed as 1.0 (*n* = 4 for each). (**B**) Protein expression of NOX2 in M_2_-MAC protein lysates. Blots were probed with anti-NOX2 (approx. 60 kDa) and anti-ACTB (approx. 45 kDa) antibodies. (**C**–**E**) Real-time PCR examination of IL-8 (**C**), IL-10 (**D**), and CEBPB (**E**) in vehicle- and NOX4/NOX2 inhibitor GLX351322 (GLX) (10 and 100 μM)-treated M_2_-MACs for 12 h. The mRNA expression level in the vehicle control is expressed as 1.0 (*n* = 4 for each). (**F**,**G**) Quantitative detection of IL-8 (**F**) and IL-10 (**G**) secretion by an ELISA assay in vehicle- and GLX (100 μM)-treated M_2_-MACs for 24 h. The cytokine secretion level in the vehicle control is expressed as 1.0 (*n* = 4 for each). (**H**–**J**) Real-time PCR examination of IL-8 (**H**), IL-10 (**I**), and CEBPB (**J**) in vehicle- and selective NOX2 inhibitor GSK2796039 (GSK) (10 μM)-treated M_2_-MACs for 12 h. The mRNA expression level in the vehicle control is expressed as 1.0 (*n* = 4 for each). (**K**,**L**) Quantitative detection of IL-8 (**K**) and IL-10 (**L**) secretion by an ELISA assay in vehicle- and GSK-treated M_2_-MACs for 24 h. The cytokine secretion level in the vehicle control is expressed as 1.0 (*n* = 4 for each). **: *p* < 0.01 vs. the vehicle control.

**Figure 9 ijms-25-09612-f009:**
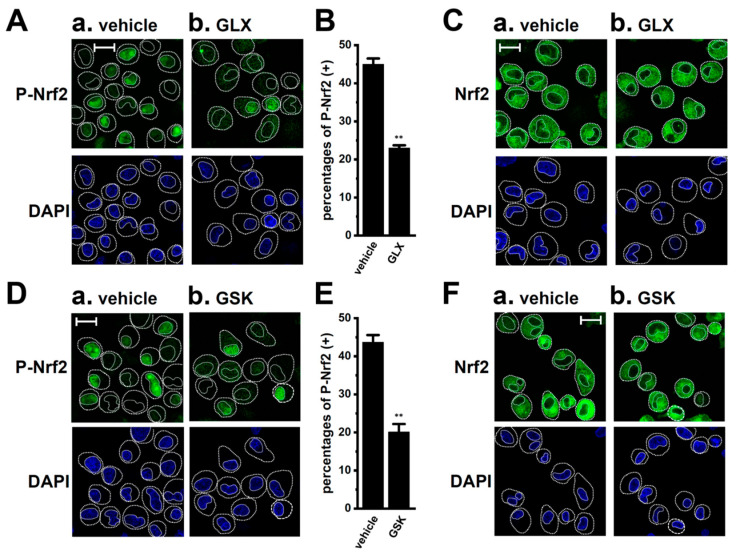
Suppressive effects of the nuclear translocation of P-Nrf2 by NOX2 inhibition in M_2_-MACs. (**A**,**D**) Confocal fluorescent images of Alexa Fluor 488-labeled P-Nrf2 (upper panels, single green image) in nuclei in vehicle- (**A**(a), **D**(a)), GLX351322 (GLX, 100 μM) (**A**(b))-, and GSK2795039 (GSK, 10 μM) (**D**(b))-treated M_2_-MACs for 2 h. (**B**,**E**) Summarized results of the percentages of P-Nrf2-positive [P-Nrf2(+)] M_2_-MACs treated with GLX (**B**) and GSK (**E**) in ‘A’ and ‘D’ (*n* = 6 for each). In each batch (*n* = 1), more than 30 cells were observed by confocal laser scanning microscopy. **: *p* < 0.01 vs. the vehicle control. (**C**,**F**) Confocal fluorescent images of Alexa Fluor 488-labeled Nrf2 (upper panels, single green image) in vehicle (**C**(a),**F**(a))-, GLX (**C**(b))-, and GSK (**F**(b))-treated M_2_-MACs for 2 h. Nuclear morphologies are shown by DAPI images (lower panels). Thick and thin dashed lines show the plasma membrane and nuclear boundary, respectively. The scale bars in ‘**A**(a)’ (same in **A**(b), ‘**C**(a)’ (same in **C**(b)), ‘**D**(a)’ (same in **D**(b)), and ‘**F**(a)’ (same in **F**(b)) show 20 μm.

**Figure 10 ijms-25-09612-f010:**
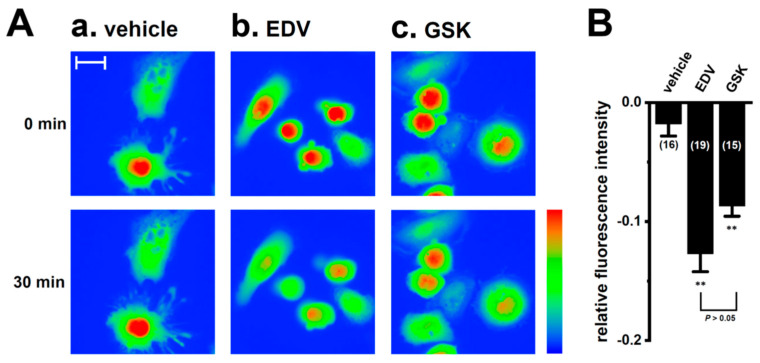
Inhibition of ROS activity by treatments with EDV and GSK in M_2_-MACs. (**A**) Typical pseudo-colored images of the fluorescent compound 2,7-dichlorofluorescein oxidized by ROS 30 min after the treatments with vehicle (a), endovion (EDV, 1 μM) (b), and GSK2795039 (GSK, 10 μM) (c) in M_2_-MACs. The color bar is scaled in the same range, and the warmer colors represent higher ROS levels. (**B**) Summarized results of relative fluorescence intensity (1.0 at time 0 min) were obtained from the optical density of 2,7-dichlorofluorescein. Numbers used for experiments are shown in parentheses. The scale bar in ‘**A**(a)’ (same in **A**(b,c)) shows 20 μm. **: *p* < 0.01 vs. the vehicle control.

**Figure 11 ijms-25-09612-f011:**
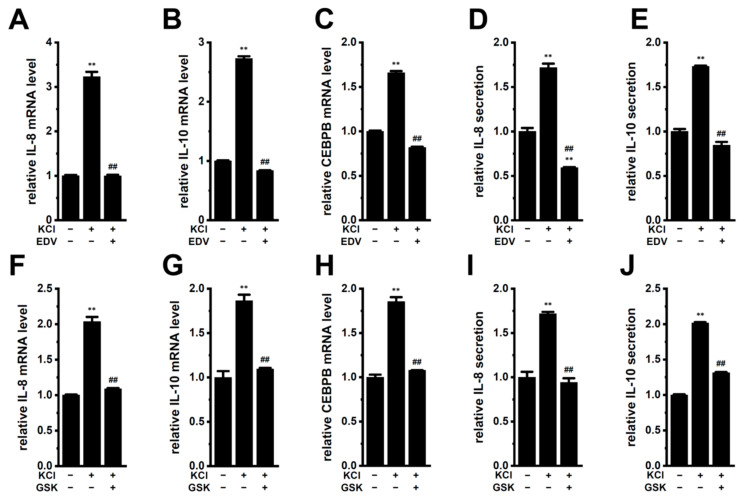
Effects of EDV on high [K^+^]_e_-stimulated IL-8 and IL-10 expression and secretion in M_2_-MACs. (**A**–**C**) Real-time PCR examination of IL-8 (**A**), IL-10 (**B**), and CEBPB (**C**) expression in normal [K^+^]_e_ (5 mM) (−)- and high [K^+^]_e_ (35 mM) (+)-treated M_2_-MACs for 24 h in the presence (+) or absence (−) of endovion (EDV; 10 μM) (*n* = 4 for each). (**D**,**E**) Quantitative detection of IL-8 (**D**) and IL-10 (**E**) secretion by an ELISA assay in vehicle- and EDV-treated M_2_-MACs for 24 h (*n* = 4 for each). (**F**–**H**) Real-time PCR examination of IL-8 (**F**), IL-10 (**G**), and CEBPB (**H**) expression in normal [K^+^]_e_ (5 mM) (−)- and high [K^+^]_e_ (35 mM) (+)-treated M_2_-MACs for 24 h in the presence (+) or absence (−) of GSK2796039 (GSK) (10 μM) (*n* = 4 for each). (**I**,**J**) Quantitative detection of IL-8 (**I**) and IL-10 (**J**) secretion by an ELISA assay in vehicle- and GSK-treated M_2_-MACs for 24 h (*n* = 4 for each). Both mRNA expression and cytokine secretion in normal [K^+^]_e_ (−/−) are expressed as 1.0. **,^##^: *p* < 0.01 vs. −/− and +/−, respectively.

**Figure 12 ijms-25-09612-f012:**
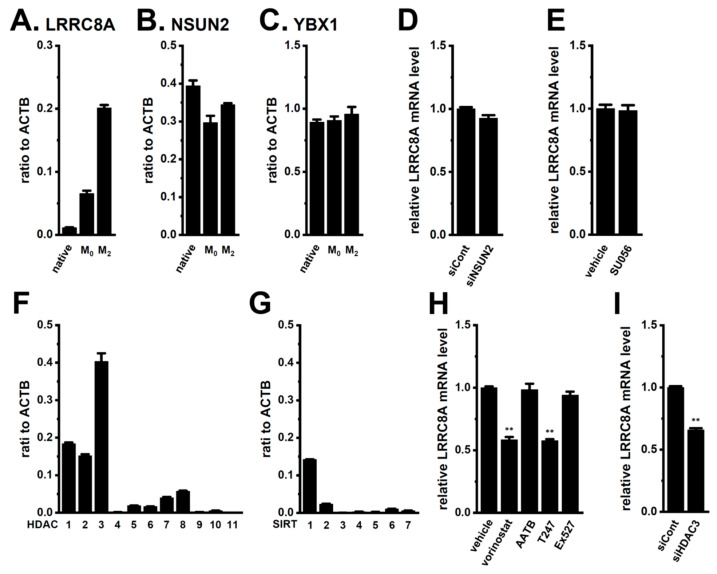
HDAC3-mediated transcriptional repression of LRRC8A in M_2_-MACs. (**A**–**C**) Real-time PCR examination of LRRC8A (**A**), NSUN2 (**B**), and YBX1 (**C**) in native THP-1 and M_0_- and M_2_-MACs (*n* = 4 for each). Expression levels are shown as the ratio to the ACTB. (**D**,**E**) Effects of the siRNA-mediated NSUN2 inhibition (siNSUN2) (**D**) and pharmacological inhibition of YBX-1 with SU056 (10 μM) for 12 h (**E**) on the expression levels of LRRC8A transcripts in M_2_-MACs (*n* = 4 for each). (**F**,**G**) Identification of histone deacetylase (HDAC) (**F**) and sirtuin (SIRT) (**G**) isoforms expressed in M_2_-MACs by a real-time PCR examination. Expression levels are shown as the ratio to the ACTB (*n* = 4 for each). (**H**) Effects of treatments with the pan-HDAC inhibitor, vorinostat (1 μM), the HDAC1/2 dual inhibitor, AATB (10 μM), the selective HDAC3 inhibitor, T247 (10 μM), and the selective SIRT1 inhibitor, Ex527 (1 μM), for 24 h on the expression levels of LRRC8A transcripts in M_2_-MACs. (**I**) Effects of siRNA-mediated HDAC3 inhibition (siHDAC3) on the LRRC8A expression levels in M_2_-MACs. mRNA expression levels in siCont and the vehicle control are expressed as 1.0 (*n* = 4 for each). **: *p* < 0.01 vs. the vehicle control or siCont.

## Data Availability

The original contributions presented in this study are included in the article/Appendix A; further inquiries can be directed to the corresponding author.

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
