# Peer review of "Downregulation of IL-8 and IL-10 by LRRC8A Inhibition through the NOX2–Nrf2–CEBPB Transcriptional Axis in THP-1-Derived M2 Macrophages"

_ijms, 2024, doi:10.3390/ijms25179612_

Round 1

Reviewer 1 Report

Comments and Suggestions for Authors

In this manuscript authors examined the role of LRRC8A in IL-8 and IL-10 expression in THP-1-differentiated M2-like macrophages (M2-MACs). In M2-MACs, the pharmacological inhibition of LRRC8A led to hyperpolarizing responses after a transient depolarization phase. Moreover, both the siRNA-mediated and pharmacological inhibition of LRRC8A repressed IL-8 and IL-10, secretion. The inhibition of LRRC8A decreased the nuclear translocation of phosphorylated NRF2, while the activation of NRF2 reversed the LRRC8A inhibition-induced transcriptional repression of IL-8 and IL-10 in M2-MACs. Authors also identified CEBPB as a down-stream target of NRF2 signaling in M2-MACs.

The manuscript and topic are interesting. The manuscript is generally well written but several points deserve to be improved. See my comments below. 

Lines 71-79: NRF2/KEAP1 signaling is poorly introduced but it plays a key role in this manuscript. The multifaceted role of this signaling deserves to be highlighted since it plays a key role in several cancerous and non-cancerous diseases (see PMID: 37525922 and 39034715 ). 

Figures must be shown early after they are mentioned in the text

Figure 1C: Increase images size 

When immunofluorescence images are shown, merged channels must always be shown 

Figure 9: Images are too small to appreciate the staining and morphology

Authors must report the product code of all kits and reagents used to ensure the reproducibility of the experiments

4.4. Real-time PCR assay: The primers used should be moved in a dedicate table 

4.5. Western blots: The primary antibodies used should be moved in a dedicate table. Moreover, antibodies product codes must be added

Add the number of replicates (N) in the legend of each figure

Although mentioned in the manuscript, supplementary figures are not present 

Abbreviations must be written in full length when mentioned for the first time

Author Response

We would like to thank the reviewer for his/her valuable comments. We have attended to all the points raised by the reviewers. Each comment is highlighted below with our response underneath.

1. Lines 71-79: NRF2/KEAP1 signaling is poorly introduced but it plays a key role in this manuscript. The multifaceted role of this signaling deserves to be highlighted since it plays a key role in several cancerous and non-cancerous diseases (see PMID: 37525922 and 39034715). 

In agreement with the reviewer’s comments, we added the description on Nrf2-Keap1 signaling (Lines 71-75). Thank you for your careful reading and telling us the related references from recent studies.

2. Figures must be shown early after they are mentioned in the text

We prepared the submitted version according to the IJMS submission form. After acceptance of the paper, the figures will be placed in appropriate positions by the assistant editor.

3. Figure 1C: Increase images size 

According to the reviewer’s suggestion, the images in Figure 1C were magnified.

4. When immunofluorescence images are shown, merged channels must always be shown 

The location of the cell membrane and nucleus are indicated by dotted lines from the transmission image, but the images were not merged in Figures 5, 9, S5, S7, and S8. We added the description in the figure legends.

5. Figure 9: Images are too small to appreciate the staining and morphology

In the confocal imaging experiments, we isolated cells with trypsin solution and were then seeded onto the glass bottom dishes. Therefore, we do not assess the morphological changes by drug application in this study. We are sorry for misleading you. We added the process of cell isolation and re-seeding onto the glass bottom dishes in Sections 4.3., 4.6. and 4.7. (immunocytochemical experiments, membrane potential and [Ca2+]i measurement, and ROS imaging) (Lines 614-616, 654-655, and 665-668).

From Figures 9A and 9D, we calculated the percentages of P-Nrf2-positive cells, and the data were summarized in Fig. 9B and 9E. Magnifying the images makes it difficult to estimate the positive cell rate. Therefore, we did not change the images in Figure 9.

Additionally, differentiated M2-like macrophages are not uniform in morphology (See below, transmission images). (See Word file)

6. Authors must report the product code of all kits and reagents used to ensure the reproducibility of the experiments

According to the reviewer’s suggestion, we added the product codes of all kits and reagents in Section 4.1.

7. 4.4. Real-time PCR assay: The primers used should be moved in a dedicate table 

According to the reviewer’s suggestion, we moved the information on PCR primers used in this study to Supplementary Table S1.

8. 4.5. Western blots: The primary antibodies used should be moved in a dedicate table. Moreover, antibodies product codes must be added

According to the reviewer’s suggestion, we moved the information on antibodies used in this study to Supplementary Table S2.

9. Add the number of replicates (N) in the legend of each figure

According to the reviewer’s suggestion, we added the number of replicates in all figure legends.

10. Although mentioned in the manuscript, supplementary figures are not present.

After receiving the reviewers’ comments, we asked the assistant editor to send the Supplementary Figure file to the reviewers. The assistant editor sent it on Aug 23.

11. Abbreviations must be written in full length when mentioned for the first time

We thank the reviewer for careful reading of our manuscript. We all checked the abbreviations again and amended the wrong ones.

Reviewer 2 Report

Comments and Suggestions for Authors

The manuscript by Matsui et al investigates the role of LRRC8A in the expression of IL-8 and IL-10 in THP-1-differentiated M2-like macrophages (M2-MACs).

A substantial part of the experimental analyses of the signaling pathways regulating IL-8 and IL-10 expression is due to the use of pharmacological inhibitors of LRRC8A with Endovion (EDV), the AKT inhibitor AZD5363, NOX4/NOX2 inhibitor GLX351322 (GLX), NOX2 inhibitor GSK2796039 (GSK), as well as HDAC inhibitors, and a SIRT1 inhibitor. From the data, the authors conclude that LRRC8A inhibition through the NOX2-Nrf2-CEBPB transcriptional axis leads to downregulation of IL-8 and IL-10 in the THP-1-derived macrophages studied.

The study is experimentally very well performed. For the correct interpretation of the data, however, it is necessary to show corresponding toxicity tests for the pharmacological inhibitors used (at the respective concentrations used). Figure 10, for example, shows that the use of inhibitors, in this case EDV and GSK, leads to morphological changes in the cells. It can therefore not be ruled out that the expression and/or activation changes of IL-8, IL-10 and the analyzed signaling factors are caused by toxic processes, which would jeopardize the message of the paper. Corresponding toxicity assays could provide the necessary clarity here.

Author Response

We would like to thank the reviewer for his/her valuable comments. We have attended to all the points raised by the reviewers. Each comment is highlighted below with our response underneath.

1. It is necessary to show corresponding toxicity tests for the pharmacological inhibitors used (at the respective concentrations used). Figure 10, for example, shows that the use of inhibitors, in this case EDV and GSK, leads to morphological changes in the cells. It can therefore not be ruled out that the expression and/or activation changes of IL-8, IL-10 and the analyzed signaling factors are caused by toxic processes, which would jeopardize the message of the paper. Corresponding toxicity assays could provide the necessary clarity here.

(1) In the confocal imaging experiments in Sections 4.3., 4.6. and 4.7. (immunocytochemical experiments, membrane potential and [Ca2+]i measurement, and ROS imaging), we isolated cells with trypsin solution and were then seeded onto the glass bottom dishes. Therefore, we do not assess the morphological changes by drug application in this study. In the experiments in Figure 10, adherent cells were isolated by the treatment with trypsin solution, and isolated cells were seeded onto the glass bottom dish for 4-6 hr. We are sorry for misleading you. We added the process of cell isolation and re-seeding onto the glass bottom dishes in Sections 4. (Lines 614-616, 654-655, and 665-668). As shown in Figure 10, 30 minutes after drug treatments little morphological changes by drug application were observed. Additionally, differentiated M2-like macrophages are not uniform in morphology (see below, transmission images). Therefore, the differential morphology of M2-MACs in Figure 10 is not due to the drug treatment.

(See Word file)

(2) We agree with the reviewer’s comment that the data on drug toxicity are needed. Indeed, the reagents used in this study include drugs with anti-cell proliferative effects. Therefore, we examined the WST cell viability assay. As shown below, no significant changes were found under the conditions of concentrations and incubation times used in this experiment.

(See Word file)

Round 2

Reviewer 1 Report

Comments and Suggestions for Authors

the manuscript has been significantly improved and can be accepted in the present form